# CLAA: Cross-Layer Attention Aggregation for Accelerating LLM Prefill

**Bradley McDanel** [1 2]   **Steven Li** [2]   **Harshit Khaitan** [2]

## Abstract

Token-ranking heuristics accelerate the prefill bottleneck in long-context LLM inference by selectively processing semantically relevant tokens. However, current evaluation relies on end-to-end benchmarks, making it difficult to isolate the quality of the token ranking itself. To address this, we introduce an Answer-Informed Reference framework with two variants: a Model Reference (Model-Ref) that measures token importance using the model generated answer, and a Ground-Truth Reference (GT-Ref) that uses human reference answers. Using GT-Ref, we establish that as few as 10% of prompt tokens suffice to match full-context performance on LongBench across all three evaluated models. The framework also reveals that existing heuristics exhibit variance across layers, with rankings degrading sharply at specific layers. Motivated by this instability, we propose Cross-Layer Attention Aggregation (CLAA), which aggregates importance scores across consecutive layers, eliminating the layer-dependent accuracy collapse observed in single-layer methods. A meaningful gap remains between the best heuristic and GT-Ref, indicating theoretical room for improved token selection.

## 1. Introduction

The prefill stage of long-context LLM inference scales quadratically with sequence length, making it a computational bottleneck for inputs of tens or hundreds of thousands of tokens. One emerging framework to reduce this cost is to operate on a smaller subset of only the most important prompt tokens. For instance, when processing a long document, many of the tokens can be irrelevant to the posed query.

The specific heuristics employed within this framework vary in both their ranking signals and their architectural designs. For ranking, importance scores are derived from different sources: methods like GemFilter (Shi et al., 2024) and FastKV (Jo et al., 2026) derive their ranking signal from the final tokens of the input prompt, while Speculative Prefill (Liu et al., 2025) uses a separate smaller model to draft a plausible future continuation, which then serves as the basis for scoring the prompt. Architecturally, GemFilter restarts computation from scratch on only the selected tokens, whereas FastKV dynamically prunes tokens during the forward pass. This variety makes principled comparison difficult, as current benchmarks rely on aggregate end-to-end performance that can mask critical failure modes, such as a method that scores well overall but systematically drops tokens at certain document positions or model layers.

To address these challenges, we propose an Answer-Informed Reference framework (Figure 1) for evaluating token ranking heuristics. Our approach is founded on the observation that the importance of a prompt token can be measured by the attention it receives from the generated answer. Because each answer token naturally attends to the prompt tokens that informed its generation, this provides a task-specific importance signal grounded in the reasoning process of the model itself. We instantiate this as a Model-Ref, which uses the answer generated by the model, and a GT-Ref, which uses human reference answers. By comparing any heuristic ranking against these references, we can directly measure how well a method identifies the tokens that matter for a given query, independent of architectural choices.

Using this framework, we find that existing heuristics exhibit significant ranking instability across model layers, with performance depending heavily on which layer is chosen for scoring. This motivates Cross-Layer Attention Aggregation (CLAA), which stabilizes rankings by aggregating importance scores across consecutive layers, producing higher similarity to the reference. In summary, our main contributions are:

- We introduce an Answer-Informed Reference that measures token importance by aggregating attention from the generated answer back to the prompt.

- Using ground-truth reference answers, we establish

[1]Franklin and Marshall College [2]Meta Reality Labs. Correspondence to: Bradley McDanel <bmcdanel@fandm.edu>.

*Proceedings of the 43rd International Conference on Machine Learning*, Seoul, South Korea. PMLR 306, 2026. Copyright 2026 by the author(s).

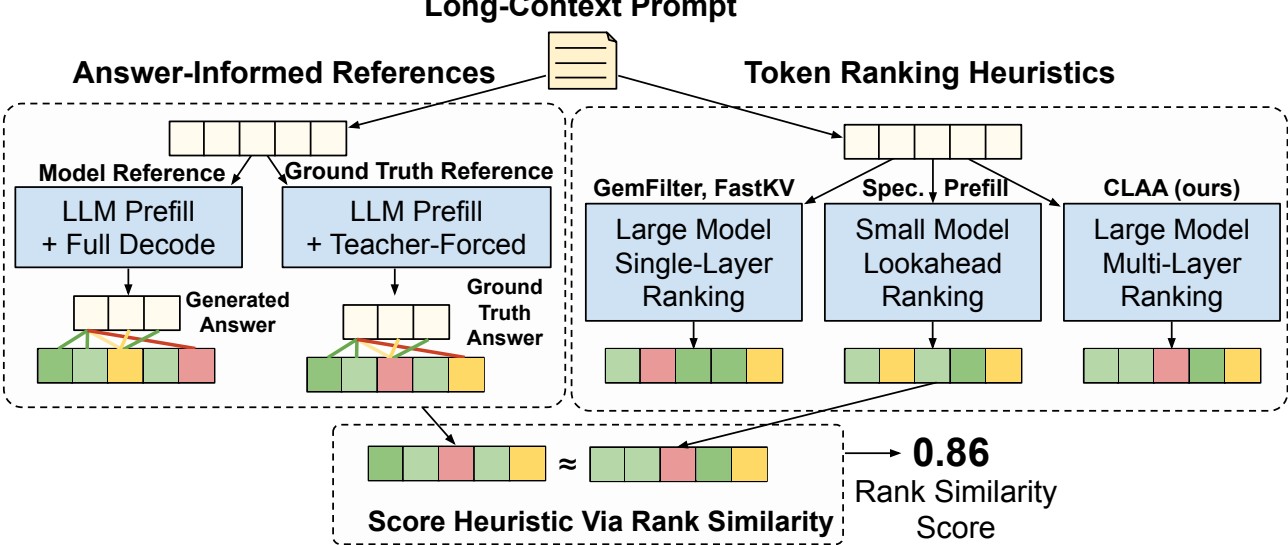

*Figure 1.* Illustration of our framework for evaluating token ranking heuristics for LLM Prefill acceleration. An Answer-Informed Reference establishes a token ranking by aggregating attention from the generated answer back to the prompt. This approach, which measures rank similarity between heuristic outputs and the reference, motivates our Cross-Layer Attention Aggregation (CLAA) method that achieves higher alignment with the reference.

that as few as **10% of prompt tokens are sufficient** to match full-context performance on LongBench.

- We propose Cross-Layer Attention Aggregation (CLAA), which addresses known layer instability (Jo et al., 2026) by aggregating rankings across consecutive layers rather than selecting a single layer.

**Conflict of Interest Disclosure.** All authors are affiliated with Meta, which develops the Llama models evaluated in this paper.

## 2. Related Work

### 2.1. Approaches to Prefill Acceleration

Hardware-aware optimizations such as FlashAttention (Dao et al., 2022; Dao, 2024) achieve substantial speedups through improved memory access patterns, though they still maintain quadratic complexity with respect to sequence length. Researchers have developed dynamic sparse attention methods to address this fundamental scaling limitation. Methods like SpAtten (Wang et al., 2021), H2O (Zhang et al., 2023), and MInference (Jiang et al., 2024) use the observation that attention patterns vary dynamically with input content, adaptively identifying important tokens at runtime rather than relying on static sparsity patterns.

Alternative architectural solutions offer different trade-offs. State space models like Mamba (Gu & Dao, 2023) achieve linear complexity but face challenges with precise token recall, which has led to the development of hybrid archi-

tectures such as Jamba (Lieber et al., 2024). Semantic compression techniques including LLMLingua (Jiang et al., 2023; Pan et al., 2024) employ smaller auxiliary models to rewrite lengthy prompts into more concise representations, achieving substantial compression ratios but introducing computational overhead from the compression process itself. Architectures like YOCO (Sun et al., 2024) require additional training or fine-tuning phases.

Token-ranking heuristics take a different approach. These methods are training-free and reduce computational costs by identifying and processing only the most relevant token subsets. They can be applied directly to existing models without modification. This paper presents an evaluation framework and improved heuristic for such approaches.

### 2.2. Token Ranking Strategies in Prefill Acceleration

Several recent methods accelerate the prefill stage by avoiding a full-sequence forward pass through all layers. These approaches share a common strategy of first ranking the importance of prompt tokens using an initial, less computationally expensive ranking heuristic. This ranking allows the model to perform its main, compute-intensive forward pass on only a top-ranked subset of tokens, thereby lowering the total prefill cost. While these heuristics all rely on attention scores to perform this ranking, their primary distinction lies in the choice of query vectors (Q) used to probe the key vectors (K) of the prompt. To clarify these differences, we formalize each approach below using a consistent notation.

Let $T_{\text{prompt}}$ be an input prompt of length $L$. For a given

model $M$ and layer $l$, let $K_{\text{prompt}}^{(l,h)} \in \mathbb{R}^{L \times d_k}$ be the matrix of key vectors for all prompt tokens at head $h$, where $d_k$ is the head dimensionality. The goal of each strategy is to compute an importance score $S_i$ for each prompt token $i \in \{1, \dots, L\}$.

**GemFilter.** GemFilter (Shi et al., 2024) hypothesizes that the query from the *very last token* of the prompt, after being processed by some initial layers, is sufficient to identify relevant context. It runs the model $M$ for $r$ layers to produce the query vector for the last token, $\mathbf{q}_{\text{last}}^{(r)}$. The importance score for the $i$-th prompt token is then computed by summing the raw attention scores (pre-softmax) from this single query across all attention heads $h$:

$$S_i^{\text{GF}} = \sum_h \left[ \frac{\mathbf{q}_{\text{last}}^{(r,h)}(K_{\text{prompt}}^{(r,h)})^\top}{\sqrt{d_k}} \right]_i \tag{1}$$

**FastKV.** FastKV (Jo et al., 2026) uses a small observation window, $\mathcal{W}$, consisting of the $W$ most recent prompt tokens as queries. This allows a collective assessment from multiple positions at the end of the context. These queries are selected from a specific Token-Selective Propagation (TSP) layer, denoted $l_{\text{TSP}}$. The importance score is derived by summing the post-softmax attention probabilities from each query in the window across all attention heads $h$:

$$S_i^{\text{FKV}} = \sum_{j \in \mathcal{W}} \sum_h \left[ \text{Softmax} \left( \frac{\mathbf{q}_j^{(l_{\text{TSP}},h)}(K_{\text{prompt}}^{(l_{\text{TSP}},h)})^\top}{\sqrt{d_k}} \right) \right]_i \tag{2}$$

FastKV also observes that critical token overlap stabilizes in later layers, using this to motivate TSP layer placement. Our reference framework provides an additional perspective on this instability, measuring it via rank correlation rather than downstream accuracy (Section 4.1).

**Speculative Prefill.** In contrast, Speculative Prefill (Liu et al., 2025) uses a separate, smaller *speculator model*, $M_{\text{spec}}$, to look into the "future." It generates $k$ lookahead tokens and uses their corresponding query vectors, $\{\mathbf{q}_{\text{gen},j}\}_{j=1}^k$, to score the prompt. This assesses token importance based on what a model *would* look for while generating a plausible continuation. The final score is the mean of the maximum raw attention scores (pre-softmax) from each lookahead query, taken across all layers and heads:

$$S_i^{\text{SP}} = \frac{1}{k} \sum_{j=1}^k \left( \max_{l,h} \left[ \frac{\mathbf{q}_{\text{gen},j}^{(l,h)}(K_{\text{prompt, spec}}^{(l,h)})^\top}{\sqrt{d_k}} \right]_i \right) \tag{3}$$

where $K_{\text{prompt, spec}}^{(l,h)}$ are the prompt key vectors as computed by the speculator model $M_{\text{spec}}$.

## 2.3. KV Cache Management in Prefill Acceleration

In addition to pruning tokens during prefill, ranking heuristics are also used to compress the KV cache for decode.

**Compression via Sequence Pruning.** For methods like GemFilter and Speculative Prefill, KV cache compression is a direct consequence of their architectural design. These heuristics first identify a single, shared subset of $L_{\text{pruned}}$ important tokens. They then execute a second, main forward pass using only this pruned sequence. The KV cache is therefore built exclusively for these $L_{\text{pruned}}$ tokens. For any given layer $l$, the resulting key cache matrix, $K^{(l)} \in \mathbb{R}^{L_{\text{pruned}} \times d_k}$, is therefore uniform across all attention heads.

**Layer-wise Cache Compression.** In contrast, FastKV employs a dual-strategy architecture that performs explicit compression of the KV cache at each layer. At each layer $l$ before the final pruning step, it computes an importance score, $\mathcal{I}$, for each key-value head group $g$:

$$\mathcal{I}_{i,g}^{\text{KV-FKV}} = \frac{1}{|\mathcal{H}_g|} \sum_{h \in \mathcal{H}_g} \sum_{j \in \mathcal{W}} \left[ \text{Softmax} \left( \frac{\mathbf{q}_j^{(l,h)}(K_{\text{prompt}}^{(l,g)})^\top}{\sqrt{d_k}} \right) \right]_i \tag{4}$$

Based on these scores, a compressed KV cache is stored independently for each head group. Critically, while the stored cache is compressed, the full, unpruned hidden states are propagated to the subsequent layer for computation. This separation allows full-context processing with a memory-efficient cache.

**Decode-time Cache Compression.** SnapKV (Li et al., 2024) takes a related approach to KV cache compression but operates at a different stage. Like FastKV, it scores prompt tokens using attention from an observation window of recent tokens, with 1D pooling to cluster nearby positions. However, SnapKV runs after full prefill and compresses the stored KV cache independently per attention head, reducing memory during decode rather than reducing computation during prefill. Because all layers process the full sequence before compression, SnapKV does not reduce Time-to-First-Token. The methods in this paper are complementary: token-ranking heuristics prune the sequence during prefill to reduce TTFT, while SnapKV-style compression can be applied to the remaining cache for further decode-time savings.

## 3. Answer-Informed Reference Framework

### 3.1. Ranking Tokens by Answer Attention

In this section, we define a reference ranking based on the full generated answer in order to compare the token rankings produced by the heuristics in Section 2.2 against a common

**Algorithm 1** Answer-Informed Reference Token Ranking

---

**Require:** Model $M$, prompt tokens $T_{\text{prompt}}$
**Require:** Answer set $\{T_{\text{ans}}^1, \dots, T_{\text{ans}}^K\}$

1: ▷ *Extract key vectors from the full prompt*
2: $K_{\text{prompt}} \leftarrow \text{GETKEYS}(\text{MODELFORWARD}(M, T_{\text{prompt}}))$

3: **for** $k = 1$ to $K$ **do**
4:      ▷ *Process answer $k$ and extract query vectors*
5:      $S \leftarrow \text{MODELFORWARD}(M, T_{\text{prompt}} \oplus T_{\text{ans}}^k)$
6:      $Q \leftarrow \text{GETQUERIES}(S, T_{\text{ans}}^k)$

7:      ▷ *Score each prompt token by attention from answer queries*
8:      $A^k \leftarrow \text{COMPUTEATTN}(Q, K_{\text{prompt}})$
9:      $S_{\text{ref}}^k \leftarrow \text{POOL1D}(\text{MEANAGG}(\text{MAXAGG}(A^k)))$
10: **end for**

11: ▷ *Average rankings across all answers*
12: $S_{\text{ref}} \leftarrow \frac{1}{K} \sum_k S_{\text{ref}}^k$
13: **return** $S_{\text{ref}}$

---

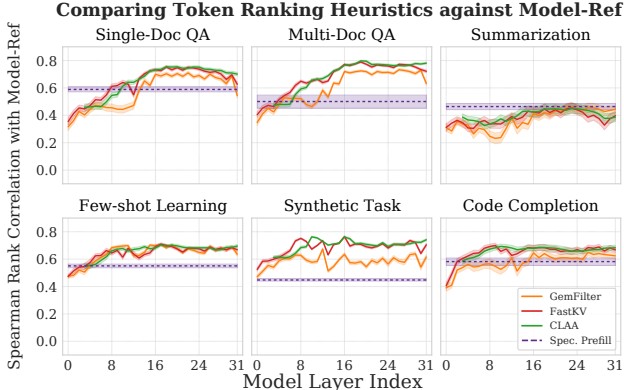

*Figure 2.* Layer-wise token ranking performance on Llama-3.1-8B-Instruct. Spearman correlation with the answer-informed reference across LongBench tasks, comparing existing heuristics to our proposed CLAA method.

baseline. For each prompt token, we compute the attention it receives across all answer tokens and layers, producing an importance score that reflects which tokens the model actually used during generation. Because this requires a completed answer, the reference cannot be used at inference time. It instead serves two roles: as a baseline for measuring ranking correlation with heuristic methods, and as an upper bound on task performance at a given token keep rate.

The reference ranking is computed by comparing prompt key vectors against answer query vectors across all layers and heads (Algorithm 1). Given a prompt and a set of answer sequences, the model extracts prompt keys once, then for each answer performs a forward pass over the concatenated prompt and answer to obtain query vectors at the answer positions. Attention scores between answer queries and prompt keys are aggregated per prompt token by taking the maximum across layers and heads, averaging across answer tokens, and applying 1D average pooling to reduce local noise. To measure how well a heuristic's ranking aligns with the reference, we use Spearman Rank Correlation ($\rho$), which compares the ordinal rankings of all prompt tokens and ranges from $-1$ (perfect disagreement) to 1 (perfect agreement).

### 3.2. Answer Sources

We analyze two approaches to obtaining the answer sequences used to compute rankings in Algorithm 1. The Model Reference (Model-Ref) produces the answer by greedy decoding from the model itself. Model-Ref requires no external labels and can be computed for any prompt, making it the default variant for evaluating heuristics throughout this paper.

The Ground-Truth Reference (GT-Ref) uses human reference answers from the evaluation dataset. Each reference answer is teacher-forced through the model in a single for-

ward pass rather than generated autoregressively, so the resulting queries reflect internal representations conditioned on a correct answer rather than on a potentially incorrect generation. When a dataset provides multiple references, averaging across them reduces variance from any single answer.

### 3.3. Reference Evaluation

Beyond measuring ranking correlation, the reference can also establish an empirical upper bound on task performance at any given token keep rate by selecting the top-ranked tokens and running inference with only those tokens. To ensure a fair comparison, the reference-guided forward pass must precisely mirror the architectural strategy of the evaluated heuristic. For methods like GemFilter and FastKV that prune at an intermediate layer $l_p$, the reference emulation mirrors this two-stage structure. In layers before $l_p$, pre-computed reference rankings compress the KV cache by selecting top-ranked token pairs, while full hidden states propagate forward to preserve the computational path. At the pruning layer $l_p$, these same rankings perform sequence pruning, filtering the hidden states themselves to retain only top-ranked tokens. This reduced sequence then flows through all subsequent layers, with the KV cache at $l_p$ compressed using the same indices.

## 4. Cross-Layer Attention Aggregation (CLAA)

### 4.1. Layer-wise Ranking Instability

Figure 2 reveals two limitations of existing token-ranking methods when examined layer by layer against our reference baseline. First, heuristics such as FastKV and GemFilter exhibit high variance, with rank correlations fluctuating dramatically and experiencing sharp drops at specific layers. This volatility is risky for methods that depend on a single

layer to determine sequence pruning decisions. When the selected layer (e.g., $l_{\text{TSP}}$) coincides with one of these performance troughs, the resulting token rankings become unreliable, potentially eliminating important tokens and compromising downstream task performance.

Second, our analysis reveals that early layers consistently show unreliable token rankings. The rank correlations in initial layers (particularly layers 0-4) are lower than those in deeper layers. Pruning based on early-layer rankings uses low-fidelity signals. Consequently, essential key-value pairs may be prematurely discarded, undermining the ability of the model to construct accurate semantic representations in subsequent layers and degrading the contextual information available for generation. As we confirm experimentally in Section 5.3, this instability translates directly into downstream accuracy loss when the pruning layer coincides with one of these troughs.

### 4.2. Multi-Layer Aggregation for Robust Ranking

To address this instability, we propose Cross-Layer Attention Aggregation (CLAA), which aggregates scores across consecutive layers. First, given the unreliability of initial layers for ranking decisions, CLAA defers KV cache compression during the first $m = 4$ layers of the model. By maintaining the complete context in these early stages, we ensure that the model can construct high-fidelity semantic representations before any KV compression occurs. Second, rather than relying on potentially unstable single-layer rankings, CLAA aggregates token importance signals across multiple consecutive layers. By averaging importance scores across these layers, CLAA smooths out layer-specific noise, making the method robust against single-layer failures. At a designated pruning layer $l_p$, CLAA synthesizes information from a window of $n$ preceding layers, $\mathcal{L} = \{l_p - n + 1, \ldots, l_p\}$. For each prompt token $i$ and layer $l' \in \mathcal{L}$, we compute a layer-specific importance score $S_i^{(l')}$ based on attention from a small observation window $\mathcal{W}$ containing the $W$ most recent prompt tokens:

$$S_i^{(l')} = \sum_{j \in \mathcal{W}} \sum_h \left[ \text{Softmax} \left( \frac{\mathbf{q}_j^{(l',h)}(K_{\text{prompt}}^{(l',h)})^\top}{\sqrt{d_k}} \right) \right]_i \quad (5)$$

Here, $\mathbf{q}_j^{(l',h)} \in \mathbb{R}^{d_k}$ represents the query vector from the $j$-th token in the observation window at layer $l'$ and head $h$, while $K_{\text{prompt}}^{(l',h)} \in \mathbb{R}^{L \times d_k}$ contains key vectors for all $L$ prompt tokens.

When the model reaches the pruning layer $l_p$, CLAA computes the final importance score for each token by averaging across all collected layer scores: $S_i^{\text{CLAA}} = \frac{1}{|\mathcal{L}|} \sum_{l' \in \mathcal{L}} S_i^{(l')}$. This mean aggregation strategy smooths out layer-specific

noise, making the ranking robust against single-layer failures.

## 5. Experiments

### 5.1. Experimental Setup

**Models and Datasets.** We evaluate our approach on Llama-3.2-3B-Instruct (28 layers) (Grattafiori et al., 2024), Llama-3.1-8B-Instruct (32 layers), and Mistral-Nemo-12B-Instruct (40 layers) (Mistral AI Team, 2024). We evaluate token ranking quality using three benchmarks: (1) Long-Bench (Bai et al., 2024) covers 16 English tasks across single/multi-document QA, summarization, few-shot learning, code completion, and synthetic tasks; (2) Needle-in-a-Haystack (Kamradt, 2023) tests information retrieval by embedding facts at various depths in 16K-64K token contexts; (3) RULER (Hsieh et al., 2024) evaluates retrieval, multi-hop tracing, information aggregation, and QA across 12 subtasks using 64K contexts with 500 samples per subtask.

**Baselines.** We compare CLAA against three recent token ranking heuristics. GemFilter (Shi et al., 2024) uses attention from the last prompt token after processing through early layers. FastKV (Jo et al., 2026) employs an observation window of recent tokens to score importance. Speculative Prefill (Liu et al., 2025) uses Llama-3.2-1B-Instruct as a draft model to generate 8 lookahead tokens for scoring; we use a 1B draft to represent resource-constrained deployment scenarios where larger drafters are not feasible. For fair comparison, all methods run on the same Hugging-Face stack with FlashAttention-2, and Speculative Prefill token rankings were verified to match the original authors' implementation. Absolute TTFT numbers differ from the original paper due to our use of HF Transformers rather than vLLM with PagedAttention; results are intended for relative comparison across methods.

To ensure an equitable comparison, a given token keep rate is applied consistently to both sequence pruning during prefill and KV cache compression for decoding across all methods. For both FastKV and CLAA, the observation window size ($W$) is set to 8 tokens. To stabilize token rankings, we apply 1D average pooling with a kernel size of 7 to the importance scores for all applicable methods. Unless otherwise specified, all layer-based ranking methods (GemFilter, FastKV, and CLAA) use layer 15 of the respective model as the pruning layer. For CLAA, we set the cross-layer aggregation window size ($n$) to 4 and keep the KV cache for the first $m = 4$ layers uncompressed. These choices are justified by ablation studies: pruning layer sensitivity is analyzed in Figure 4, cross-layer window size and first uncompressed layer in Appendix B. All hyperparameter configurations are summarized in Table 6.

*Table 1.* Llama-3.1-8B LongBench results with 10% token keep rate. FullKV is the baseline using 100% of tokens.

| Method | Single-Document QA | | | Multi-Document QA | | | Summarization | | | Few-shot Learning | | | Synthetic | | Code | | |
| | NrtvQA | Qasper | MF-en | HotpotQA | 2WikiMQA | MuSiQue | GovReport | QMSum | MultiNews | TREC | TriviaQA | SAMSum | PCount | PRe | LCC | RB-P | Avg. |
|---|---|---|---|---|---|---|---|---|---|---|---|---|---|---|---|---|---|
| FullKV | 30.16 | 45.53 | 54.94 | 56.02 | 46.66 | 31.28 | 35.12 | 25.28 | 27.25 | 73.00 | 91.65 | 43.80 | 8.88 | 99.50 | 63.38 | 56.73 | 49.32 |
| *Answer-Informed References (Upper Bounds)* | | | | | | | | | | | | | | | | | |
| Model-Ref | 29.85 | 43.94 | 55.85 | 54.99 | 47.11 | 28.92 | 32.26 | 25.29 | 21.95 | 68.50 | 91.43 | 41.80 | 7.47 | 99.50 | 59.63 | 56.85 | 47.83 |
| GT-Ref | 29.74 | 47.23 | 55.20 | 54.84 | 49.17 | 31.07 | 30.91 | 26.56 | 25.55 | 73.00 | 91.87 | 44.60 | 6.44 | 99.50 | 64.21 | 58.68 | **49.29** |
| *Heuristic Methods* | | | | | | | | | | | | | | | | | |
| GemFilter | 24.36 | 21.07 | 39.73 | 51.29 | 33.92 | 25.78 | 28.94 | 18.98 | 17.42 | 60.50 | 91.53 | 40.39 | 4.76 | 87.50 | 22.73 | 32.47 | 37.59 |
| SpecPrefill | 28.53 | 32.86 | 51.94 | 54.33 | 40.80 | 29.66 | 27.47 | 22.43 | 19.76 | 62.50 | 89.31 | 40.14 | 4.40 | 66.08 | 50.49 | 51.09 | 41.99 |
| FastKV | 30.60 | 38.96 | 53.61 | 54.87 | 44.73 | 30.09 | 28.08 | 24.57 | 20.93 | 70.00 | 92.38 | 42.69 | 6.56 | 99.00 | 58.43 | 53.49 | 46.81 |
| CLAA | 31.09 | 42.36 | 53.68 | 53.83 | 44.73 | 31.53 | 28.15 | 24.76 | 20.42 | 70.00 | 92.37 | 42.93 | 6.51 | 99.50 | 58.31 | 53.86 | **47.13** |



*Figure 3.* Needle-in-a-Haystack result of Llama-3.1-8B-Instruct with 40% token keep rate. X denotes out of memory on 80GB A100.

*Table 2.* Average LongBench results across models at 10% token keep rate. SpecPrefill is not applicable to Mistral-Nemo-12B as no compatible speculator model is available.

| Method | Llama-3.2-3B | Llama-3.1-8B | Mistral-Nemo-12B |
|---|---|---|---|
| FullKV (100%) | 44.23 | 49.32 | 48.32 |
| Model-Ref | 44.13 | 47.83 | 47.75 |
| GT-Ref | **45.35** | **49.29** | **48.95** |
| GemFilter | 35.20 | 37.59 | 36.60 |
| FastKV | 42.45 | 46.81 | 45.62 |
| SpecPrefill | 34.06 | 41.99 | N/A |
| CLAA | **42.93** | **47.13** | **46.05** |

**Evaluation Details.** For task evaluation, we report task-specific metrics from LongBench (F1 for QA tasks, Rouge-L for summarization, accuracy for classification) and binary retrieval accuracy for Needle-in-a-Haystack. For the RULER benchmark, we report the official recall-based accuracy averaged across its tasks. Experiments used a single A100 GPU (80GB). For the reference, we use the same main model to generate complete answers. All settings use greedy decoding (temperature 0).

## 5.2. Main Results

Table 1 presents LongBench results on Llama-3.1-8B at 10% token keep rate. Notably, GT-Ref achieves 49.29%, matching the full-context FullKV baseline (49.32%) while using only 10% of prompt tokens. This confirms that for these tasks, the information needed to answer each query is concentrated in a small fraction of the prompt. Model-Ref, which relies on the model's own generated answer rather than human references, scores 47.83%, approximately 1.5 points below GT-Ref. Among heuristic methods, CLAA achieves 47.13%, closing the majority of the gap between FastKV (46.81%) and Model-Ref. Paired bootstrap tests (10k resamples) confirm that Model-Ref outperforms both

FastKV (+0.99, $p < 0.05$) and CLAA (+0.60, $p < 0.05$), while the CLAA–FastKV difference (+0.39) is not significant on most individual datasets; the primary advantage of CLAA is layer stability rather than aggregate accuracy at a fixed layer.

CLAA also demonstrates greater robustness than single-layer methods in retrieval tasks. The Needle-in-a-Haystack results (Figure 3) reveal critical weaknesses in baseline methods: GemFilter fails at intermediate document positions (22–44% depth), while SpecPrefill misses needles in the latter half. CLAA maintains consistent accuracy across all positions (0.909 average). RULER (Table 3) confirms this pattern at 64K context, with CLAA leading in retrieval (89.85%) and multi-hop reasoning (87.72%). These results align with the reference analysis: tasks requiring distributed facts benefit from multi-layer aggregation, as a token ignored at one layer may be deemed important at another.

*Table 3.* RULER benchmark performance by category with 64K context on Llama-3.1-8B with 40% token keep rate.

| Method | Retr. | M-Hop | Agg. | QA | Avg. |
|--------|-------|-------|------|------|------|
| FullKV | 98.27 | 86.88 | 86.67 | 63.30 | 83.78 |
| GemFilter | 82.23 | 58.20 | **88.33** | 63.70 | 73.12 |
| FastKV | 87.69 | 87.16 | 86.67 | 63.60 | 81.28 |
| CLAA | **89.85** | **87.72** | 86.67 | **63.80** | 82.01 |

*Table 4.* Spearman correlation with GT-Ref and downstream accuracy around the FastKV collapse region at 10% keep rate (six-task average). The correlation dip corresponds directly to the accuracy collapse.

| Model | Layer | FastKV | | CLAA | |
|-------|-------|--------|------|------|------|
| | | Corr. | Acc. | Corr. | Acc. |
| Llama 3.2-3B | 10 | 0.537 | 41.40 | 0.526 | 42.06 |
| | 11 | 0.468 | 35.05 | 0.528 | 42.45 |
| | 12 | 0.580 | 46.76 | 0.583 | 46.62 |
| Llama 3.1-8B | 11 | 0.603 | 50.87 | 0.593 | 51.50 |
| | 12 | 0.546 | 41.80 | 0.603 | 52.54 |
| | 13 | 0.618 | 53.29 | 0.635 | 52.97 |

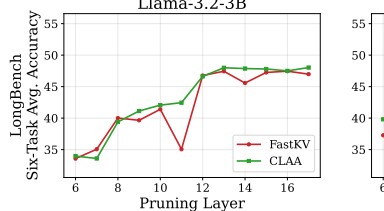
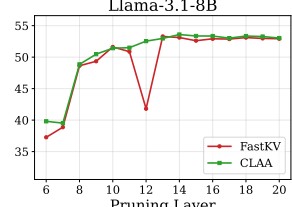

*Figure 4.* Downstream accuracy of FastKV and CLAA as a function of pruning layer at 10% token keep rate, averaged over six LongBench tasks (NarrativeQA, Qasper, MultifieldQA-en, MuSiQue, TREC, TriviaQA). FastKV collapses at layer 11 for Llama-3.2-3B and layer 12 for Llama-3.1-8B, while CLAA remains stable across all layers.

## 5.3. Pruning Layer Sensitivity

Figure 4 sweeps the pruning layer across both Llama models at 10% token keep rate. FastKV drops by 12 points at layer 11 on Llama-3.2-3B (35.05 vs. 47.25 at layer 15) and by 11 points at layer 12 on Llama-3.1-8B (41.80 vs. 53.29 at layer 13). The collapse layer differs between models, so no single layer choice is universally safe. CLAA remains stable through both collapse regions because it aggregates scores across multiple layers, preventing any single unreliable layer from dominating the ranking.

Table 4 connects ranking quality to downstream performance across both models. In each case, the layer where FastKV Spearman correlation with GT-Ref is lowest (layer 11 for 3B, layer 12 for 8B) is also the layer where FastKV accuracy collapses. CLAA accuracy at the same layers remains stable, comparable to neighboring layers. The collapse layer differs between models and cannot be identified without sweeping, making single-layer methods inherently fragile. The selection of layer 15 for our main experiments is justified as a balance between computational savings and performance, as accuracy plateaus beyond that point for both methods. For example, FastKV at layer 19 scores 47.20 on Llama-3.1-8B, comparable to CLAA at layer 15 (47.13), but TTFT increases from 2054 ms to 2489 ms because layers 16–18 must process the full sequence. Additional ablations on the aggregation window size $n$ and the first compression layer $m$ are provided in Appendix B.

## 5.4. Reference Gap Analysis

Heuristic methods cannot match the reference upper bound because they lack information. The reference has access to attention patterns from the full generated answer, whereas heuristics must estimate token importance from the prompt alone, before generation begins. This information gap sets a ceiling on prefill compression: no prompt-only ranker can recover the attention signal carried by the generated answer, and no generated answer fully matches the signal carried by the human reference. The two resulting margins bound the achievable error. The reference also separates ranking quality from task difficulty. If a method scores poorly, it is unclear whether the low score reflects poor ranking quality or inherent task difficulty. The reference establishes a baseline, bounding the minimum error achievable with perfect token importance information.

We expect this gap to vary by task structure. When the relevant tokens (e.g., keywords or entity names) are predictable from the prompt alone, heuristics should approach the reference upper bound. When the answer path depends on subtle contextual cues, the gap should widen. Token importance is not a static property of the prompt but depends on which output the model produces.

Our results confirm this pattern. On TriviaQA, heuristics nearly match Model-Ref: CLAA achieves 92.37% versus the reference's 91.43% at 10% keep rate, within noise. The query explicitly names the target entity, so heuristics identify relevant tokens without knowing the answer. In contrast, tasks like Qasper show larger gaps (CLAA 42.36% vs. Model-Ref 43.94%), suggesting the relevant context is harder to predict from the query alone.

The reference framework also clarifies the value of lookahead. Speculative Prefill generates tokens with a small model before ranking, partially closing the information gap. However, it still lags behind simpler heuristics: at 10% keep rate, Speculative Prefill achieves 41.99% versus CLAA's

47.13% (Table 2). The 1B speculator poorly predicts which tokens the 8B model will attend to, limiting the benefit of lookahead.

### 5.5. Efficiency and Accuracy Trade-off

Figure 5 shows the accuracy-speed trade-off for each method. CLAA tracks the reference ceiling more closely than single-layer methods, consistent with the diagnosis that layer instability was the limiting factor. At a 10% keep rate, CLAA reduces TTFT by 39% compared to FullKV (from roughly 900ms to 550ms). It achieves 47.13% accuracy, closely approaching Model-Ref at 47.83%. FastKV offers the next best performance. In contrast, GemFilter and Speculative Prefill exhibit a larger accuracy degradation for a similar reduction in TTFT. Figure 6 provides a detailed breakdown of prefill and decode performance. CLAA leaves the first four layers uncompressed to maintain foundational token representations before aggregation. This results in a minimal increase in KV cache size, for example, 0.3 GB for CLAA versus 0.1 GB for Model-Ref at a 10% keep rate. This modest increase in memory is justified by accuracy improvements (Figure 5), showing a favorable trade-off between resource usage and performance. In contrast, the GemFilter approach results in a larger KV cache (1.3 GB at a 10% keep rate) because its implementation retains the full cache and indexes into it during decoding, which negatively impacts decode throughput (16 tps compared to 19-20 tps for others).

We also compare against semantic compression methods, which use auxiliary models to rewrite prompts into concise representations before processing with the main model. We evaluate two representative methods: Selective Context (Li et al., 2023) and LLMLingua (Jiang et al., 2023). Both use Llama-3.2-1B-Instruct as the compression model and Llama-3.1-8B-Instruct as the main model.

Table 5 reports end-to-end TTFT including the compression overhead. The autoregressive rewrite step dominates total latency for both semantic compression methods. Selective Context is slower than the FullKV baseline on all three tasks due to its 3–5 second rewrite overhead, while also degrading task quality. LLMLingua achieves modest speedups on longer contexts (GovReport) but still underperforms CLAA on both speed and quality. In contrast, CLAA achieves 1.34x–1.45x speedup over FullKV while maintaining accuracy within 4% of the baseline. Semantic compression methods could benefit in amortized settings where the same rewritten prompt is reused across multiple queries; however, for single-query inference, token-ranking heuristics like CLAA offer a more favorable efficiency-quality trade-off.

*Table 5.* Comparison with semantic compression methods on Llama-3.1-8B. Rewrite time is the compression model overhead; Total TTFT includes both rewrite and prefill.

| Task | Method | Rewrite | Total TTFT | Speedup | Score |
|------|--------|---------|-----------|---------|-------|
| Qasper | FullKV | — | 1084 ms | 1.00x | 0.487 |
| | Selective Context | 3134 ms | 3421 ms | 0.32x | 0.186 |
| | LLMLingua | 901 ms | 1141 ms | 0.95x | 0.129 |
| | CLAA (20%) | — | 755 ms | **1.44x** | **0.470** |
| GovReport | FullKV | — | 13505 ms | 1.00x | 0.377 |
| | Selective Context | 5642 ms | 15391 ms | 0.88x | 0.327 |
| | LLMLingua | 1815 ms | 9888 ms | 1.37x | 0.236 |
| | CLAA (20%) | — | 10055 ms | **1.34x** | **0.310** |
| TriviaQA | FullKV | — | 1768 ms | 1.00x | 0.944 |
| | Selective Context | 5254 ms | 6027 ms | 0.29x | 0.204 |
| | LLMLingua | 1682 ms | 2412 ms | 0.73x | 0.470 |
| | CLAA (20%) | — | 1217 ms | **1.45x** | **0.944** |

## 6. Limitations and Future Work

While CLAA achieves strong performance across retrieval and QA tasks, several limitations warrant discussion. First, summarization tasks such as Multi-News exhibit relatively flat rank correlations for all methods, including CLAA (Figure 2). In these tasks, token importance appears to evolve dynamically throughout generation rather than being fixed at the prompt boundary. Static prefill pruning, which commits to a single importance ranking before any tokens are generated, may be fundamentally limited for such workloads. Future methods may need to re-assess token importance dynamically during decoding to capture these shifts.

Second, our evaluation focuses on single-turn inference. In multi-turn conversation settings, token pruning methods face an additional challenge: importance estimates from the first user turn may not remain valid as the conversation evolves. While these methods could theoretically be re-run for each additional turn, doing so would eventually negate the efficiency gains. Extending token-ranking heuristics to multi-turn settings remains an open problem shared by all methods in this class.

Finally, all methods in our evaluation, including CLAA, use FlashAttention-2 for the main forward pass. The scoring computation required by CLAA (a $W \times L$ attention computation per layer in the aggregation window) accounts for less than 2% of total TTFT and is already included in our reported wall-clock times. No modifications to underlying CUDA kernels are required.

The accuracy gap between GT-Ref and the best heuristic indicates that current token selection methods leave significant headroom. Because GT-Ref rankings are derived from ground-truth answers, they provide a natural supervision signal for learning token importance predictors. A lightweight model trained to approximate GT-Ref rankings from the prompt alone could narrow this gap without requir-

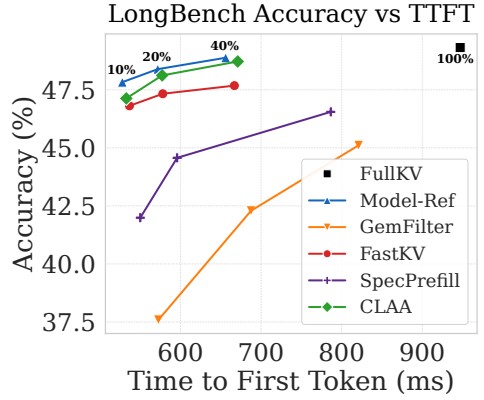

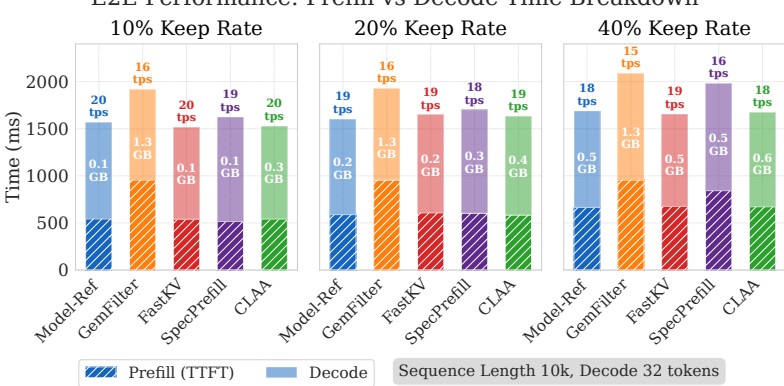

*Figure 5.* LongBench accuracy versus Time-to-First-Token (TTFT) for Llama-3.1-8B-Instruct on a 10k token sequence. Points correspond to 10%, 20%, and 40% keep rates.

*Figure 6.* End-to-end performance breakdown for a 10k token prompt and 32 token generation. Bars show Prefill (TTFT) and Decode time. Annotations indicate decode throughput (tokens per second) and KV cache size (GB) at the start of decode.

ing answer-time attention at inference. More broadly, the reference framework enables rapid evaluation of token selection strategies by measuring ranking quality directly, decoupling scoring architecture design from full downstream benchmarking. Extending the framework to quantify how token importance shifts during generation would clarify which workloads require dynamic re-ranking and which are well served by static prefill pruning.

## 7. Conclusion

We show that token-ranking heuristics for prefill acceleration can be evaluated directly against an Answer-Informed Reference, decoupling ranking quality from architectural design choices. This reveals two key findings: first, as few as 10% of prompt tokens are sufficient to match full-context performance when the right tokens are selected; second, existing heuristics suffer from layer-wise ranking instability. CLAA addresses this instability through multi-layer aggregation, producing more stable rankings without additional model parameters or training.

A meaningful gap between the best heuristic and the ground-truth reference remains across all evaluated models, suggesting that current methods have not saturated the potential of token-ranking approaches. The reference framework provides both a benchmark for measuring progress and a natural supervision signal for learning token importance predictors.

## Acknowledgements

We thank the anonymous reviewers for their constructive feedback. In particular, reviewer suggestions to validate our reference framework against ground-truth answers led to the GT-Ref experiments, which substantially strengthened the paper.

## Impact Statement

This paper presents work whose goal is to advance the field of machine learning. There are many potential societal consequences of our work, none of which we feel must be specifically highlighted here.

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

## A. Hyperparameter Configuration

This appendix details the hyperparameter configurations for all methods to ensure a transparent and reproducible comparison. Key architectural parameters, such as the pruning layer index, were swept across all applicable methods for our ablation studies (Section 5.3). The final values used in our main results (e.g., Tables 1 and 2) were selected to provide a consistent and fair comparison point across methods.

## B. CLAA Hyperparameter Ablations

In our main experiments, we set the number of initial uncompressed layers to $m = 4$ and the aggregation window size to $n = 4$. Figure 7 (left) shows that starting compression from layer 0 results in the lowest accuracy, as the model must build initial representations from compressed context. Delaying compression until layer 4 provides an accuracy boost, justifying $m = 4$ as a default. Figure 7 (right) shows that expanding from $n = 1$ to $n = 2$ provides an accuracy boost at all keep rates, validating that cross-layer aggregation mitigates single-layer instability. Performance stabilizes around $n = 4$, which we use throughout.

*Table 6.* Hyperparameter settings for all methods evaluated on Llama-3.1-8B-Instruct.

| Method | Parameter | Symbol | Value(s) Explored | Final Value in Main Results |
|---|---|---|---|---|
| **Model-Ref** | Token Keep Rate | - | {0.1, 0.2, 0.4} | Main Variable |
| (Ours) | Pruning Layer Index | $l_p$ | {3, 7, 11, 15, 19} | 15 |
| **FastKV** | Token Keep Rate | - | {0.1, 0.2, 0.4} | Main Variable |
| | TSP Layer Index | $l_{\text{TSP}}$ | {3, 7, 11, 15, 19} | 15 |
| | Observation Window | $W$ | {8} | 8 |
| | Pooling Kernel Size | - | {7} | 7 |
| **GemFilter** | Token Keep Rate | - | {0.1, 0.2, 0.4} | Main Variable |
| | Routing Layer Index | $r$ | {3, 7, 11, 15, 19} | 15 |
| **SpecPrefill** | Token Keep Rate | - | {0.1, 0.2, 0.4} | Main Variable |
| | Lookahead Tokens | $k$ | {8} | 8 |
| **CLAA** | Token Keep Rate | - | {0.1, 0.2, 0.4} | Main Variable |
| (Ours) | Pruning Layer Index | $l_p$ | {3, 7, 11, 15, 19} | 15 |
| | Aggregation Window | $n$ | {1, 2, 4, 8, 12} | 4 |
| | First Uncompressed Layer | $m$ | {0, 2, 4, 6} | 4 |
| | Observation Window | $W$ | {8} | 8 |
| | Pooling Kernel Size | - | {7} | 7 |

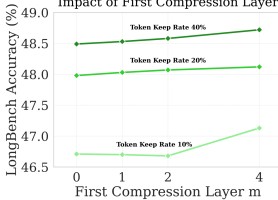
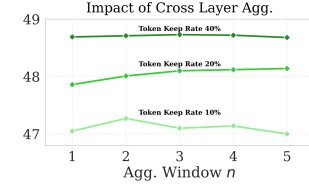

*Figure 7.* **Left**: Impact of the first compression layer index ($m$) on LongBench accuracy for CLAA. Deferring compression to later layers (increasing $m$) improves accuracy. **Right**: Impact of the cross-layer aggregation window size ($n$) at different token keep rates. Expanding from $n = 1$ to $n = 2$ provides an accuracy boost, and performance stabilizes around $n = 4$. Both experiments use Llama-3.1-8B-Instruct.

## C. Per-Task Ranking Correlation Analysis

This section analyzes token ranking performance by comparing various heuristics against the Answer-Informed Reference. Figure 8 shows Spearman Rank Correlation for each heuristic across all model layers.

- **Layer-specific volatility:** Single-layer heuristics show performance drops at specific layers. In NarrativeQA, GemFilter and FastKV exhibit sharp correlation drops around layer 10. This pattern repeats across QASPER, HotpotQA, and QMSum. CLAA avoids these drops through multi-layer aggregation.

- **Practical implications:** Methods using single pre-selected layers risk severe performance degradation if their chosen layer coincides with a performance trough. CLAA provides more reliable token selection by aggregating across a layer window.

- **Performance convergence:** While layer-dependent heuristics reach similar high performance in deep layers (20-31), CLAA achieves comparable results without the intermediate volatility.

- **Universal patterns:** Early layers (0-8) consistently fail to predict token importance across all methods. Speculative Prefill serves as a task-dependent baseline with varying competitiveness.

These results demonstrate that CLAA provides more stable token ranking compared to single-layer methods, making it a robust choice for prefill acceleration.

## D. Pseudocode

This section provides unified pseudocode for all methods discussed in the paper, organized into token ranking heuristics (Section D.1) and answer-informed references (Section D.2).

### D.1. Token Ranking Heuristics

**GemFilter.** GemFilter operates in a two-pass system. It first runs a partial forward pass to rank tokens using the query from the final prompt token. It then discards this intermediate state and executes a new, standard forward pass using only the top-ranked tokens and their original positions (Listing 1).

**FastKV.** Unlike GemFilter's two-pass system, FastKV operates in a single, continuous forward pass. At each layer, it uses an "observation window" of recent tokens to

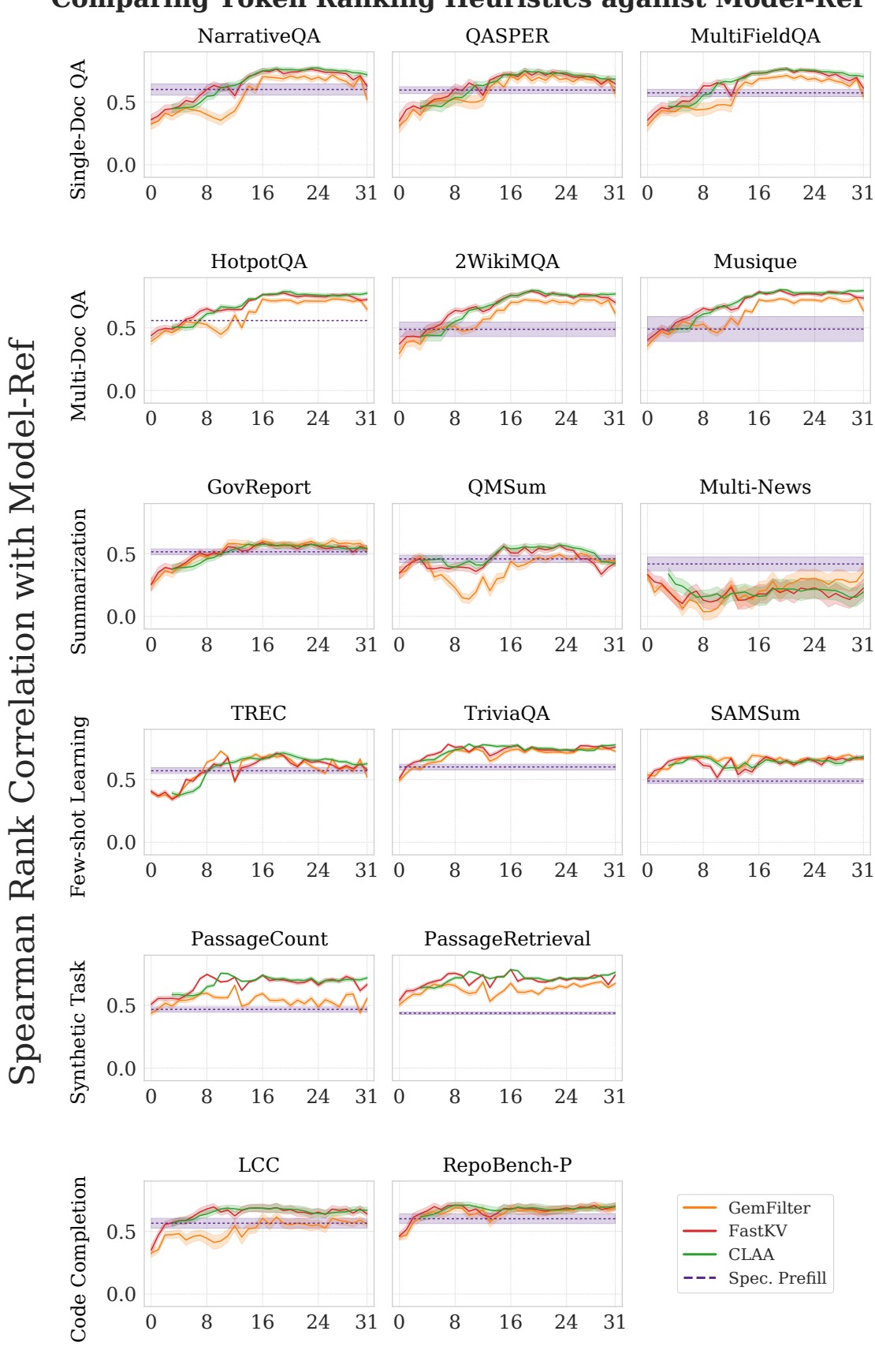

*Figure 8.* Detailed per-task comparison of token ranking heuristics against the Answer-Informed Reference.

rank all prompt tokens and compress the stored KV cache. At a designated Token-Selective Propagation (TSP) layer, it performs a one-time sequence pruning, after which the forward pass continues on the reduced sequence (Listing 2).

**Speculative Prefill.** Speculative Prefill uses a two-model architecture. A small "speculator" model first performs a full prefill and generates a few lookahead tokens. The queries from these generated tokens are used to rank the importance of the original prompt tokens. Finally, the large "base" model performs a selective prefill on only the top-ranked prompt tokens, using their original position IDs to maintain context (Listing 3).

**CLAA.** Our proposed method enhances the single-pass architecture by improving ranking stability. It first defers any compression for an initial $m$ layers. Then, for each subsequent layer, it computes importance scores using an observation window and stores them in a buffer of size $n$. At a designated pruning layer, $l_p$, it aggregates scores across this buffer by averaging across layers and taking the maximum across heads (Listing 7).

### D.2. Answer-Informed References

Both reference variants share a common evaluation phase that performs a selective prefill using pre-computed token rankings (Listing 6). They differ in how rankings are obtained.

**Model-Ref.** The Model Reference generates an answer autoregressively, then measures the attention from each generated token back to the prompt to compute token importance scores (Listing 4). This does not require external labels and can be computed for any prompt.

**GT-Ref.** The Ground-Truth Reference teacher-forces human reference answers through the model in a single forward pass, producing attention-based importance scores conditioned on a correct answer rather than a potentially incorrect generation (Listing 5). When multiple reference answers are available, rankings are averaged across them.

```python
def gemfilter_prefill(model, prompt_tokens, routing_layer, keep_rate):
    # Run a partial forward pass to get the query from the last prompt token
    hidden_states = model.forward(prompt_tokens, stop_at_layer=routing_layer)
    q_last = hidden_states.get_last_token_query(layer=routing_layer)
    k_all = hidden_states.get_keys(layer=routing_layer)

    # Rank tokens by their raw attention (pre-softmax) to the last token q
    scores = aggregate_attention(q_last, k_all, use_softmax=False)
    top_k_indices = topk(scores, keep_rate).indices

    pruned_tokens = gather(prompt_tokens, top_k_indices)

    # Restore position ids
    original_position_ids = gather(range(len(prompt_tokens)), top_k_indices)

    # Execute a full forward pass on pruned sequence
    final_outputs, final_kv_cache = model.forward(
        pruned_tokens,
        position_ids=original_position_ids
    )

    return final_outputs, final_kv_cache
```

*Listing 1.* GemFilter Prefill Logic.

```python
def fastkv_prefill(model, prompt_tokens, tsp_layer, window_size, keep_rate):
    hidden_state = model.embed(prompt_tokens)
    kv_cache = {}

    for l in range(model.num_layers):
        # Get queries from the observation window
        q_window = hidden_state.get_last_n_queries(n=window_size, layer=l)
        k_all = hidden_state.get_keys(layer=l)
        v_all = hidden_state.get_values(layer=l)

        # Rank tokens using post-softmax attention from the window
        scores = aggregate_attention(q_window, k_all, use_softmax=True)
        top_k_indices = topk(scores, keep_rate).indices

        # Compress KV cache using rankings (for decode)
        kv_cache[l] = gather(k_all, v_all, on_indices=top_k_indices)

        hidden_state = model.layer_forward(l, hidden_state, use_kv=(k_all, v_all))

        # At the TSP layer, prune hidden state with rankings
        if l == tsp_layer:
            hidden_state = gather(hidden_state, on_indices=top_k_indices)

    final_outputs = model.final_norm(hidden_state)
    return final_outputs, kv_cache
```

*Listing 2.* FastKV Prefill Logic.

```python
def speculative_prefill(base_model, spec_model, prompt_tokens, look_ahead_k, keep_rate):
    # Run a small speculator model on the full prompt
    spec_outputs, spec_kv_cache = spec_model.forward(prompt_tokens)
    spec_k_prompt = spec_kv_cache.get_all_keys()

    # Generate 'look_ahead_k' tokens with the speculator to get qs
    q_generated = []
    next_token = spec_outputs.get_next_token()
    for _ in range(look_ahead_k):
        lookahead_out, spec_kv_cache = spec_model.forward(next_token, spec_kv_cache)
        q_generated.append(lookahead_out.get_query())
        next_token = lookahead_out.get_next_token()

    # Rank prompt tokens using draft qs
    scores = aggregate_attention(q_generated, spec_k_prompt)
    top_k_indices = topk(scores, keep_rate).indices

    pruned_tokens = gather(prompt_tokens, top_k_indices)
    original_position_ids = gather(range(len(prompt_tokens)), top_k_indices)
    final_outputs, final_kv_cache = base_model.forward(
        pruned_tokens,
        position_ids=original_position_ids
    )
    return final_outputs, final_kv_cache
```

*Listing 3.* Speculative Prefill Logic.

```python
def get_model_ref_ranking(model, prompt_tokens, max_gen_len):
    _, prefill_kv_cache = model.forward(prompt_tokens)
    k_prompt = prefill_kv_cache.get_all_keys()

    q_generated = []
    next_token = prefill_kv_cache.get_next_token()
    current_kv_cache = prefill_kv_cache
    while not is_eos(next_token) and len(q_generated) < max_gen_len:
        output, current_kv_cache = model.forward(next_token, past_kv=current_kv_cache)
        q_generated.append(output.get_query())
        next_token = output.get_next_token()

    scores = aggregate_attention(q_generated, k_prompt, use_softmax=False)
    return scores
```

*Listing 4.* Model-Ref: Token ranking via autoregressive generation.

```python
def get_gt_ref_ranking(model, prompt_tokens, answer_tokens_list):
    _, prefill_kv_cache = model.forward(prompt_tokens)
    k_prompt = prefill_kv_cache.get_all_keys()

    all_scores = []
    for answer_tokens in answer_tokens_list:
        # Teacher-force the reference answer in a single forward pass
        combined = concatenate(prompt_tokens, answer_tokens)
        output, _ = model.forward(combined)
        q_answer = output.get_queries(answer_positions=answer_tokens)

        scores = aggregate_attention(q_answer, k_prompt, use_softmax=False)
        all_scores.append(scores)

    return mean(all_scores)
```

*Listing 5.* GT-Ref: Token ranking via teacher-forced reference answers.

```python
def reference_prefill(model, prompt_tokens, reference_scores, keep_rate):
    top_k_indices = topk(reference_scores, keep_rate).indices

    pruned_tokens = gather(prompt_tokens, top_k_indices)
    original_position_ids = gather(range(len(prompt_tokens)), top_k_indices)

    final_outputs, final_kv_cache = model.forward(
        pruned_tokens,
        position_ids=original_position_ids
    )
    return final_outputs, final_kv_cache
```

*Listing 6.* Shared reference evaluation: selective prefill using pre-computed rankings.

```python
def claa_prefill(model, prompt_tokens, pruning_layer, aggregation_window,
                 defer_layers, window_size, keep_rate):
    hidden_state = model.embed(prompt_tokens)
    kv_cache = {}
    layer_scores_buffer = collections.deque(maxlen=aggregation_window)

    for l in range(model.num_layers):
        q_window = hidden_state.get_last_n_queries(n=window_size, layer=l)
        k_all, v_all = hidden_state.get_keys(layer=l), hidden_state.get_values(layer=l)

        # Compute and buffer importance scores
        current_layer_scores = aggregate_attention(q_window, k_all, use_softmax=True)
        layer_scores_buffer.append(current_layer_scores)

        # Defer KV compression for the first 'defer_layers'
        if l < defer_layers:
            kv_cache[l] = (k_all, v_all)
        else:
            compression_indices = topk(current_layer_scores, keep_rate).indices
            kv_cache[l] = gather(k_all, v_all, on_indices=compression_indices)

        hidden_state = model.layer_forward(l, hidden_state, use_kv=(k_all, v_all))

        # At the pruning layer, aggregate and prune
        if l == pruning_layer:
            agg = mean(stack(layer_scores_buffer), dim=layers)
            final_scores = max(agg, dim=heads)
            pruning_indices = topk(final_scores, keep_rate).indices
            hidden_state = gather(hidden_state, on_indices=pruning_indices)

    final_outputs = model.final_norm(hidden_state)
    return final_outputs, kv_cache
```

*Listing 7.* Cross-Layer Attention Aggregation (CLAA) Prefill Logic.

