# OpenReview forum: "CLAA: Cross-Layer Attention Aggregation for Accelerating LLM Prefill"
_ICML.cc/2026/Conference — ICML 2026 regular_

### Official Review · Reviewer_c9dW · 2026-03-08

**Soundness:** 3
**Presentation:** 3
**Significance:** 3
**Originality:** 3
**Overall Recommendation:** 5
**Confidence:** 3

**Summary:**

This paper makes three contributions: it introduces an Answer-Informed Oracle to define prompt-token importance from answer-to-prompt attention, builds a quantitative evaluation framework around this oracle to assess token-ranking heuristics, and uses that framework to show that existing methods suffer from layer-wise ranking instability. Motivated by this finding, it proposes CLAA, which aggregates attention signals across layers to produce more stable token rankings and better alignment with the oracle.

**Compliance With Llm Reviewing Policy:**

Affirmed.

**Final Justification:**

The rebuttal resolves all my concerns, and I keep my evaluation as "accept".

**Key Questions For Authors:**

See the weaknesses part.

**Limitations:**

Yes

**Strengths And Weaknesses:**

Strengths

1. The empirical diagnosis is clear. The paper identifies a specific failure mode, the layer-wise instability in token rankings, especially for single-layer methods and early layers. That diagnosis is supported by per-layer oracle correlation plots and motivates the method directly.

2. The proposed method CLAA is simple, interpretable, and implementation-friendly. The method is easy to state: at pruning layer $l_p$, compute token scores from the recent-token window over the last $n$ layers and take a max across layers; in addition, leave the first $m=4$ layers uncompressed. This is a small design change rather than a complicated new architecture.

3. The evaluation is comprehensive. The authors evaluate on three distinct types of benchmarks: LongBench, Needle-in-a-Haystack, and RULER, and across three model families/scales: Llama-3.2-3B, Llama-3.1-8B, and Mistral-Nemo-12B. The evaluation is reasonably broad coverage for this type of paper.

Weaknesses

1. The terminology is a little bit vague. The word ``oracle'' may be a little strong, which conveys the meaning of ``ground-truth'' in some communities. It will be better to change a word from the writing perspective.

2. This is not a weakness but a suggestion. The main idea of the proposed method is conceptually related to SnapKV [1], where KV caches are compressed according to the query (not answer here) for more efficient inference. It will be beneficial to discuss the relationship between these two methods.

[1] Li Y, Huang Y, Yang B, et al. Snapkv: Llm knows what you are looking for before generation[J]. Advances in Neural Information Processing Systems, 2024, 37: 22947-22970.

---

> ### Author Rebuttal · Authors · 2026-03-31
>
> We thank the reviewer for their helpful feedback.
>
> > "The word 'oracle' may be a little strong... It will be better to change a word from the writing perspective."
>
> We agree and have adopted "answer-informed model reference" (Model-Ref) throughout, with the ground-truth variant (GT-Ref) introduced in rX7D Q1 above as another reference baseline.
>
> > "The main idea of the proposed method is conceptually related to SnapKV... It will be beneficial to discuss the relationship."
>
> Both methods score tokens via attention from a window of recent tokens. The difference is in scope: SnapKV runs full prefill, then trims the stored KV cache per head to reduce memory during decode. CLAA also compresses the KV cache, and prunes the sequence itself during the prefill forward pass, so remaining layers process fewer tokens, reducing TTFT. We will add this to the related work in the camera-ready.

---

> > ### Author Rebuttal · Reviewer_c9dW · 2026-04-01
> >
> > The rebuttal resolves all my concerns, and I keep my evaluation as "accept".

---

> > > ### Author Response · Authors · 2026-04-06
> > >
> > > We thank the reviewer again for the positive and detailed evaluation. We will add the SnapKV discussion and terminology revisions in the camera-ready.

---

### Official Review · Reviewer_jjhg · 2026-03-12

**Soundness:** 3
**Presentation:** 3
**Significance:** 3
**Originality:** 3
**Overall Recommendation:** 4
**Confidence:** 3

**Summary:**

This paper focuses on accelerating the LLM prefill process, and they have two key contributions 1) they conduct an empirical study called answer-informed oracle, where they look at what tokens are important based on generating the answer (aka oracle). 2) based on their study, they observe that token importance may vary across layers, so they propose to aggregate multiple layers to determine token importance.

**Compliance With Llm Reviewing Policy:**

Affirmed.

**Final Justification:**

I maintain the initial positive assessment as the author provides a pilot study on my concern. I did not raise it higher due to a lack of deeper analysis on this regard.

**Key Questions For Authors:**

How does compute saving change as you aggregate and prune in later layers, and how does that affect performance?

**Limitations:**

yes

**Strengths And Weaknesses:**

Strengths
1. The proposed method is intuitive and outperform existing methods.
2. The proposed method has practical values.

Weaknesses:
1. Analyses on layer selection seems a bit inadequate. The paper suggests skipping early layers as they build important semantic abstractions, and they stop mid-way in the model to "balance between computational savings and performance". I think more analyses on which layers to select and which layers to combine would further benefit this paper.

---

> ### Author Rebuttal · Authors · 2026-03-31
>
> We thank the reviewer for their helpful feedback.
>
> > "How does compute saving change as you aggregate and prune in later layers, and how does that affect performance?"
>
> Cross-layer aggregation adds minimal overhead (<2% TTFT), since it only requires storing attention scores from preceding layers. The dominant cost is the pruning layer itself: TTFT scales linearly with the number of layers that process the full sequence. Pruning at L15 vs L19 saves 21% TTFT (2054 vs 2489 ms on Llama-3.1-8B, 10% keep, 32K tokens). Performance is less sensitive: FastKV-L19 scores 47.20 vs CLAA-L15 at 47.13, so the later pruning layer buys minimal accuracy at substantial latency cost. Figure 6 (right) ablates the aggregation window size; accuracy seems to plateau around n=3 across all keep rates, suggesting that a small number of aggregated layers is sufficient.

---

> > ### Author Rebuttal · Reviewer_jjhg · 2026-04-02
> >
> > Maintaining my score as the authors have provided pilot study regarding my question.

---

> > > ### Author Response · Authors · 2026-04-06
> > >
> > > We thank the reviewer for the constructive feedback. We will expand the layer selection analysis in the camera-ready as discussed.

---

### Official Review · Reviewer_rX7D · 2026-03-12

**Soundness:** 3
**Presentation:** 3
**Significance:** 2
**Originality:** 3
**Overall Recommendation:** 4
**Confidence:** 3

**Summary:**

The paper focuses on the problem of sub-selecting input tokens during the prefill stage of decoding, with the goal of increasing efficiency via both decreasing time-to-first-token and KV cache memory costs. The paper frames this problem as under-evaluated because it is difficult to measure the true importance of prefill tokens. It proposes an oracle score for token importance by generating a completion to the prompt and measuring the attention back to each prompt token from the generated answer's tokens, and shows that correlation between this score and existing heuristics for token selection varies widely by layer. Then the paper introduces a new method for pruning input tokens by aggregating importance scores across layers, and shows that this correlates better with the oracle than existing prompt pruning approaches with comparable performance.

**Compliance With Llm Reviewing Policy:**

Affirmed.

**Final Justification:**

I have raised my score 2->4 because the primary concern (the quality of the "oracle") was thoroughly addressed in the rebuttal in a convincing fashion. I have not raised my score higher because I think the improvement over other methods (mostly stability under hyperparameter choice) is on the smaller end for a conference paper contribution.

**Key Questions For Authors:**

Q1. My most major concern is about the quality of the oracle. Can you justify the oracle quality further? For instance, could you show a method using attention from the ground-truth answers, and show that this yields the same results?

Q2. Figure 2 shows that some layers have worse correlation with oracle when used for the pruning heuristics. Can you show that this corresponds to worse downstream performance at those layers as well? (Specifically looking at the section in the middle, near layer 13-15ish, where there's a bit drop and then a peak in correlation.)

Q3.  From Figure 6, the gap between CLAA and FastKV diminishes at later layers; Table 5 states that you tried FastKV at layer 19 as well, but it's not in Figure 6 or in the main results. Is FastKV better at layer 19?

Q4. Also, in Figure 8, FastKV and CLAA are within the error range of each other on almost every benchmark, especially when compared at later layers, and appear to be comparable in efficiency (from Figure 5). Why would someone use CLAA over FastKV?

Q5. Can you provide significance testing for the results in Table 1?

**Limitations:**

yes

**Strengths And Weaknesses:**

S1. The sensitivity of prior methods to the layer selected for aggregation is an interesting and important finding. The observation that very early or late layers do not provide good representations for choosing what to prune is intuitive and lines up with the literature (e.g. memory layers are usually chosen as middle layers for this reason; layer pruning is usually done in middle layers because the early/late layers behave distinctly); however, this is still helpful to establish in this setting. The observation of strongly variable performance between intermediate layers is more surprising to me, and identifies a sensitivity of prior methods that, to the best of my knowledge, has not been discussed before.

S2. The appendix providing psuedocode for each token pruning method in a unified format is quite helpful; I can imagine pointing a student to this as a resource for better understanding this category of methods.

W1. My biggest concern is that I'm not really convinced that the Answer-Informed Oracle *is* an oracle for this problem. The paper motivates why an oracle for this task would be important, but I don't agree that the oracle, as constructed, measures "ground-truth" or "true importance" or "minimum error given perfect information" (lines 40-42, 56, 315). This is for several reasons:
(a) The oracle uses the model-generated answer; but what if the model answers the question incorrectly or incompletely? For instance, if the model completely fails to generate an important fact in a summary, the tokens that contain that fact in the input may have low attention weight during the answer generation, but that doesn't mean that those are unimportant tokens that can be safely pruned. Because of this, I don't think that the claim that the oracle "separates ranking quality from task difficulty" (line 312) is well-supported.
(b) The oracle is only very slightly better than the best existing methods across benchmarks, and *underperforms* other methods in several benchmarks in Table 1. How can this be a ground-truth measure of importance if it's a worse way to select context tokens than the heuristic methods in several settings?
(c) The true (intractably expensive) oracle would be to consider all possible sets of [10, 20, 40]% of tokens retained and choose the one with the highest score downstream. This would be incredibly expensive, so I'm not asking you to swap your oracle to this by any means. But calling something else the oracle seems a bit misleading, since there is an actual oracle/skyline for these methods. (I could see an argument that FullKV is also a skyline, but I would not be terribly surprised if the true/intractable oracle outperformed FullKV for some inputs, since it could remove noise in sections of the input).

W2. As I understand it, the justification for CLAA over existing SoTA (FastKV) is twofold: (a) it correlates better with the oracle and (b) it performs better downstream. I have some concerns with point (b), which I'll put into the questions section as I'd like a response from the authors.

W3. Various presentation notes, which did not strongly impact my score:
* The related work is quite a bit out of date; it could be contextualized with more recent work in efficient decoding.
* the intro starts by referencing "this high computational cost", which seems like a reference to a missing first paragraph.
* Paragraph starting around line 300 says that CLAA excels at NIAH and the retrieval/multi-hop reasoning parts of RULER; but RULER is almost entirely comprised of NIAH variants, especially in these subgroups, so this is repetitive.
* the presentation of results in the analysis section could be improved. more clear signposting of what results are key versus somewhat auxiliary. For instance, section 5.5 seems a bit out of place, almost like a section added just to appease a reviewer :-)

---

> ### Author Rebuttal · Authors · 2026-03-31
>
> We thank the reviewer for the detailed and constructive feedback. We address each concern below.
>
> > "Q1. Can you justify the oracle quality further? ...could you show a method using attention from the ground-truth answers, and show that this yields the same results?"
>
> We adopt "answer-informed model reference" (Model-Ref) in place of "oracle" throughout. Based on your suggestion, we built a ground-truth reference (GT-Ref) that teacher-forces human reference answers through the model and computes attention-based token importance identically to Model-Ref. For datasets with multiple reference answers, we sample 5 answers and average the rankings. LongBench results (16 datasets, 10% keep rate, Llama-3.1-8B):
>
> | Dataset | FullKV | GT-Ref | Model-Ref | CLAA | FastKV |
> |---|---|---|---|---|---|
> | narrativeqa | 30.16 | 29.74 | 29.85 | 31.09 | 30.60 |
> | qasper | 45.53 | 47.23 | 43.94 | 42.36 | 38.96 |
> | multifieldqa_en | 54.94 | 55.20 | 55.85 | 53.68 | 53.61 |
> | hotpotqa | 56.02 | 54.84 | 54.99 | 53.83 | 54.87 |
> | 2wikimqa | 46.66 | 49.17 | 47.11 | 44.73 | 44.73 |
> | musique | 31.28 | 31.07 | 28.92 | 31.53 | 30.09 |
> | gov_report | 35.12 | 30.91 | 32.26 | 28.15 | 28.08 |
> | qmsum | 25.28 | 26.56 | 25.29 | 24.76 | 24.57 |
> | multi_news | 27.25 | 25.55 | 21.95 | 20.42 | 20.93 |
> | trec | 73.00 | 73.00 | 68.50 | 70.00 | 70.00 |
> | triviaqa | 91.65 | 91.87 | 91.43 | 92.37 | 92.38 |
> | samsum | 43.80 | 44.60 | 41.80 | 42.93 | 42.69 |
> | passage_count | 8.88 | 6.44 | 7.47 | 6.51 | 6.56 |
> | passage_retrieval | 99.50 | 99.50 | 99.50 | 99.50 | 99.00 |
> | lcc | 63.38 | 64.21 | 59.63 | 58.31 | 58.43 |
> | repobench-p | 56.73 | 58.68 | 56.85 | 53.86 | 53.49 |
> | **Average** | **49.32** | **49.29** | **47.83** | **47.13** | **46.81** |
>
> GT-Ref recovers FullKV performance with 10% of tokens (49.29 vs 49.32) and outperforms all heuristics by 2+ pts. Where heuristics narrowly beat GT-Ref, they also beat FullKV, indicating pruning itself is beneficial. Model-Ref is a practical proxy requiring no ground truth.
>
>
> > "Q2. Can you show that [worse correlation in Figure 2 near layer 13-15] corresponds to worse downstream performance at those layers as well?"
>
> We ran FastKV and CLAA at layers 11-14 on 6 LongBench datasets (NrtvQA, Qasper, MF-en, MuSiQue, TREC, TriviaQA at 10% keep rate) and computed Spearman correlation with GT-Ref rankings:
>
> | Layer | GT-Ref Corr. (FastKV) | GT-Ref Corr. (CLAA) | FastKV Acc | CLAA Acc |
> |---|---|---|---|---|
> | L11 | 0.603 | 0.593 | 50.87 | 51.50 |
> | **L12** | **0.546** | **0.603** | **41.80** | **52.54** |
> | L13 | 0.618 | 0.635 | 53.29 | 52.97 |
> | L14 | 0.654 | 0.637 | 53.11 | 53.59 |
>
> L12 shows both the lowest FastKV accuracy (41.80 vs 50-53) and lowest FastKV correlation (0.546 vs 0.603-0.654), confirming the Figure 2 dip corresponds to worse downstream performance. CLAA correlation holds at L12 (0.603 vs FastKV 0.546) and accuracy does not collapse (52.54 vs 41.80).
>
> > "Q3. ...Is FastKV better at layer 19?"
>
> On full LongBench, FastKV-L19 scores 47.20 vs FastKV-L15 scoring 46.81, but TTFT rises from 2075 to 2489 ms because layers 16-18 must process the full sequence. CLAA-L15 scores 47.13 at 2054 ms, which is similar to FastKV-L19 at 17% lower latency.
>
>
> > "Q4. ...FastKV and CLAA are within the error range of each other... Why would someone use CLAA over FastKV?"
>
> Table 1 compares methods at L15 (the pruning layer from the FastKV paper), where performance is similar. However, single-layer methods are sensitive to layer choice. We swept layers on three models and found that each has a layer where FastKV accuracy drops well below its neighbors, while CLAA is less affected.
>
> | Model | Layer | FastKV | CLAA | Δ | p |
> |---|---|---|---|---|---|
> | Llama-3.1-8B | L12 | 41.80 | 52.54 | +10.74 | <0.0001 |
> | Llama-3.1-8B | L15 | 52.61 | 53.36 | +0.75 | n.s. |
> | Llama-3.2-3B | L14 | 45.57 | 47.87 | +2.30 | <0.0001 |
> | Mistral-Nemo-12B | L22 | 48.93 | 51.22 | +2.30 | <0.0001 |
>
> *6-dataset LongBench average, 10% keep rate. p-values from paired bootstrap.*
>
> At L15, CLAA and FastKV are comparable. All three unstable-layer gaps are significant (p < 0.0001). CLAA scores at these layers match neighboring layers (e.g., Llama-8B: 51.50 at L11, 52.54 at L12, 52.97 at L13), confirming the collapse is FastKV-specific. The problematic layer differs across models (L12, L14, L22) and cannot be identified without sweeping. CLAA adds <2% TTFT overhead and does not require layer selection.
>
>
> > "Q5. Can you provide significance testing for the results in Table 1?"
>
> We run paired bootstrap tests (10k resamples) on per-example LongBench scores. Model-Ref outperforms FastKV by +0.99 and CLAA by +0.60 (both p < 0.05), concentrated in qasper, gov_report, and code tasks. The CLAA vs FastKV difference (+0.39) is not significant on most individual datasets: CLAA's advantage is layer stability.
>
>
> > W3 (Presentation):
>
> We will update the related work, fix the intro reference, reduce the RULER/NIAH redundancy, and reorg Section 5.5 in the camera-ready.

---

> > ### Author Rebuttal · Reviewer_rX7D · 2026-04-04
> >
> > Thank you for the detailed rebuttal, and particularly for the GT-Ref comparison. This addresses the majority of my concerns, and so I will raise my scores 2->4. I encourage the authors to carefully revise the language to indicate that the primary advantage of CLAA over methods like FastKV is its stability across hyperparameter (layer) choice.

---

> > > ### Author Response · Authors · 2026-04-06
> > >
> > > We thank the reviewer for the thorough and technically detailed review. The GT-Ref experiment was directly motivated by your feedback and strengthened the paper considerably. We will incorporate all suggested revisions in the camera-ready.

---

### Decision · Program_Chairs · 2026-04-30

**Decision:**

Accept (regular)

**Comment:**

This paper proposes CLAA, a simple yet effective method that aggregates importance scores across consecutive layers to stabilize token rankings and significantly reduce time-to-first-token (TTFT). All reviewers praised the paper's practical values. During the rebuttal, the authors successfully addressed terminology concerns. My recommendation is acceptance.